# Improving tomato yield and production economics with Nano-NPK fertilization and salicylic acid chemigation

Zhenhao Guan[1], Khandakar Rafiq Islam[1]*, Arifur Rahman[1], Brad Bergefurd[2]

**1** Soil, Water, and Bioenergy Resources, The Ohio State University South Centers, Piketon, Ohio, United States of America, **2** Brandt Discovery & Innovations, LLC, Wilmington, Ohio, United States of America

* islam.27@osu.edu

## Abstract

Nanotechnology-enabled NPK fertilization combined with biostimulants offers a sustainable approach to enhance crop productivity, resource-use efficiency, and environmental performance in specialty crops. A two-year (2022−2023) factorial experiment (3 × 2), arranged in a completely randomized design, evaluated the interactive effects of nano humic acid–silicic acid-based Triple 20 NPK fertilizers (nano-NPK) applied at 40, 80, and 120 kg ha⁻¹, with and without 0.3% salicylic acid (SA) as biostimulant, on processing tomato (*Solanum lycopersicum* L. cv. BHN 685) grown in a low-fertility soil under drip-irrigated, raised bed plasticulture. Conventional Triple 20 NPK fertilization at 120 kg ha⁻¹ served as the control. Multivariate statistical analyses demonstrated that nano-NPK fertilization and SA, alone or in combination, significantly improved tomato yield components, water use efficiency (WUE), and fertilizer use efficiency (FUE), while reducing cull fruit and increasing marketable yield. Among treatments, 80 kg ha⁻¹ nano-NPK combined with 0.3% SA produced both total and marketable yields equivalent to or exceeding those obtained with 120 kg ha⁻¹ nano-NPK or conventional fertilization, alongside higher nutrient, and water utilization. These improvements were associated with enhanced nutrient bioavailability, uptake, and photosynthetic performance due to nano-enabled NPK fertilization, with SA further promoting plant growth and fruit quality. This combination reduced fertilizer input by up to 33% without compromising yield, achieving WUE and FUE comparable to or better than conventional NPK fertilization (120 kg ha⁻¹). Economically, 80 kg ha⁻¹ nano-NPK + 0.3% SA achieved the highest benefit–cost ratio (1.26) and net return (US $1,988 ha⁻¹), outperforming conventional NPK fertilization. Environmental assessment indicated improved energy use efficiency (4–6%) and lower greenhouse gas (GHG) intensity per unit of marketable yield. Although total GHG emissions were statistically similar at higher application rates, nano-NPK, SA, or their combination reduced GHG intensity, highlighting their sustainability advantage. Overall, integrating 80 kg ha⁻¹ nano-NPK with 0.3% SA optimizes yield, profitability, and environmental

**Data availability statement:** All relevant data are within the paper and its Supporting information file.

**Funding:** The financial support provided by the Ohio Department of Agriculture with Sponsor Award ID # AGR-SCBG-21-03.

**Competing interests:** Authors have declared that no competing interests exist.

stewardship, offering an efficient pathway for sustainable intensification of tomato production.

## Introduction

Tomato is the third most economically important crop globally after grasses and legumes, and it is the most valuable and widely cultivated specialty crop, with an annual production of approximately 182.3 million tons on about 4.85 million hectares of land [1]. Renowned for its unique culinary applications, tomato is consumed both fresh and processed products, serving as a key component of a nutritious diet and contributing to public health [2,3]. Processed tomato products, including purées, juices, ketchup, canned, and dried tomatoes, are economically valuable and continue to be used in response to increase global demand. To meet this demand, tomatoes are cultivated across a range of soils, environments, and management practices, from open fields to controlled conditions such as greenhouses and high tunnels [4].

As with other specialty crops, achieving higher tomato yields depends on the use of inputs such as chemical fertilizers, organic and inorganic amendments, organo-chemical mixtures, and natural products [5]. However, commercial fertilizers, typically applied as nitrogen (N), phosphorus (P), and potassium (K), exhibit low to moderate nutrient use efficiency (NUE), with efficiency rates of approximately 30–35% for N, 18–20% for P, and 35–40% for K [5]. These inefficiencies necessitate excessive fertilization, particularly of reactive chemical forms, to maintain productivity. This practice contributes to economic and environmental concerns, including nutrient leaching, water pollution with harmful algal blooms, greenhouse gas emissions, and the accumulation of heavy metals in soil and crops [6].

Excessive reliance on chemical fertilization not only affects agroecosystem services but also increases production costs related to energy, labor, materials, nutrients, and irrigation, reducing farm sustainability. Moreover, tomato production is increasingly challenged by abiotic and biotic stresses driven by global climate change, as well as by the growing scarcity of freshwater resources [6,7]. Improper irrigation, either insufficient or excessive, can hinder tomato growth, nutrient uptake, and overall yield. In water-scarce regions with low soil fertility, optimizing irrigation and fertilization is essential for sustainable tomato production. These challenges underscore the urgent need for innovative strategies to enhance tomato productivity while improving water-use efficiency (WUE) and NUE [7,8].

Nanotechnology offers a promising strategy for precision and improved nutrient management, aiming to enhance crop productivity through controlled and efficient nutrient delivery [9–11]. Nanomaterials, ranging in size from 1 to 100 nm, function as concentrated sources or carriers of essential or beneficial nutrients. Their high surface area-to-volume ratio and unique physicochemical properties enable effective nutrient absorption and targeted delivery to plants [12,13]. Nanopores in plant roots facilitate the efficient uptake and translocation of nano-enabled fertilizers, thereby improving NUE.

Previous studies have demonstrated that nano-enabled fertilizers, whether applied to plants independently or in combination with chemical fertilizers, significantly improve the nutrient uptake by plants to sustain yield and quality compared to the use of fertilizers alone [3,11,14]. For example, the integration of $FeO_x$ or $SiO_2$ nanoparticles with NPK fertilizers has been shown to increase tomato yield, enhance phytochemical quality, and improve WUE and NUE, especially when compared to conventional NPK formulations alone [3,15–17]. These improvements are attributed to the nanoparticles' enhanced absorption, faster reactivity, greater utilization efficiency, and reduced nutrient losses, increasing yield while minimizing environmental footprints [18,19].

Alongside nano-fertilization, salicylic acid (SA), a naturally occurring plant hormone and key stress-signaling molecule, has emerged as a promising tool to alleviate both abiotic and biotic stressors in plants [20–25]. When applied in dilute concentrations, SA enhances plant resilience to stress conditions such as drought, salinity, heavy metal toxicity, and pathogenic infections [23,25]. It does so by modulating a range of physiological and biochemical pathways, including stomatal behavior, transpiration rate, nutrient, and water uptake efficiency [20,26]. In addition, SA is known to stimulate the production of secondary metabolites like antioxidants and phytochemicals, which may contribute to improved plant vigor, fruit development, yield, and nutritional composition in vegetable crops including tomatoes [3,27–29].

To advance sustainable tomato production, it is critical to adopt an integrated evaluation framework encompassing economic viability, energy efficiency, and environmental impact [30–32]. Economic analysis plays a fundamental role in promoting the adoption of sustainable practices in managed tomato production systems [33]. Likewise, energy efficiency remains a crucial metric in assessing agricultural productivity and sustainability [34]. Enhancing energy efficiency not only conserves valuable resources but also improves farm profitability and reduces environmental burdens [35]. Among environmental concerns, greenhouse gas (GHG) emissions from inputs and agricultural systems are of global significance. Reducing GHG emissions is a strategic approach to achieving climate goals, particularly the stabilization of carbon dioxide emissions before 2030 [36].

Although several studies have addressed the economic, energy, and environmental dimensions of agricultural systems, mainly focusing on cereal crops, as well as horticultural, and other crops [33,37,38], the combined impact of nanomaterial-enhanced NPK fertilization (nano-NPK) and SA chemigation on processing tomato production has not been explored. Investigating the potential synergistic benefits of this integrated approach on yield, profitability, energy use, and environmental sustainability represents a valuable and timely research opportunity.

This study aims to address the research gap by evaluating the combined application of nano-NPK fertilizers and SA via chemigation for processing tomato cultivation in low fertility soils. The specific objectives were to: (1) assess the effects of different rates of nano-NPK fertilizers, with and without SA, on fruit yield attributes, WUE, and fertilizer use efficiency (FUE); and (2) conduct a life cycle assessment (LCA) to compare the economic viability, energy efficiency, and environmental impacts of processing tomato production under nano-NPK and SA treatments with those under conventional NPK fertilization in a drip-irrigated raised bed plasticulture system.

## Materials and methods

### Study site

A replicated field study on drip irrigated plasticulture processing tomato was conducted at South Centers research farm in Piketon, The Ohio State University, Ohio, USA (lat. 39.07° N, long. 83.01° W, elevation 103 m above sea level) during 2022–2023 growing seasons.

Long-term average monthly maximum air temperature was recorded at 32.2°C, reaching its peak in August, while the lowest temperature of 15.6°C was observed in September. The mean annual rainfall is 96.2 ± 8.3 cm, with approximately 55% occurring during the crop growing season (May to September). The highest monthly rainfall, measuring 15 cm, was recorded in July [15]. The soil is a low-fertility poorly drained Doles silt loam (*Fine-silty, mixed, active, mesic Aeric Fragiaqualf*), with a pH 6 ± 0.3, organic matter 14.1 ± 4 g kg$^{-1}$, total nitrogen 1.05 ± 0.24 g kg$^{-1}$, bulk density 1.28 ± 0.04 g cm$^{-3}$, sand, silt, and clay 300 ± 40 g kg$^{-1}$, 55 ± 20 g kg$^{-1}$, and 150 ± 20 g kg$^{-1}$, respectively, at 0–30 cm depth with fragipan [28,39].

**Experiment design and cultural practices.** A 3 x 2 factorial experiment in a completely randomized design was conducted to evaluate the effects of various levels of nano-NPK fertilization and SA chemigation on processing tomato production under plasticulture-raised bed [28]. Prior to laying the plastic mulch, the field was chisel-plowed, and a basal dose of Triple 19 fertilizer (at 20 kg ha$^{-1}$) was applied as a starter fertilizer to all plots. Triple 19 is a granular fertilizer containing 19% N, 19% P, and 19% K.

The nano-NPK fertilizer treatments consisted of Triple 20, a premium, water-soluble fertilizer with 20% N, 20% P, and 20% K, commonly used in drip irrigation systems. It was applied at equivalent rates of 20, 60, and 100 kg ha$^{-1}$, each fortified with 1% Nano Pro [a nano humic acid and silicic acid mixture], and compared to a control treatment receiving 100 kg NPK ha$^{-1}$ without Nano Pro. Additionally, treatments were combined with SA at two levels, 0% (control) and 0.3%, with four replicated plots for each treatment combination. Individual raised bed plots measure 2 m in width and 5 m in length, separated by 50 cm buffer zones [28].

Both the Triple 19 and Triple 20 NPK fertilizers were supplied by the Andersons° [https://www.andersonsinc.com]. The Nano Pro (a mixed solution of humic acid-silicic acid with adjuvant) was obtained from Aqua-Yield° Inc., Draper, Utah, USA (https://www.nano-yield.com/AquaYield], and the sodium salicylate (≥ 99% purity) used as the source of SA was obtained from Fisher Scientific° (USA).

Processing tomato seeds (cv. BHN 685) were sown in 72-cell plug trays filled with Metro Mix 360° soilless medium and placed in a controlled growth chamber for germination. The growth chamber conditions were maintained as temperatures of 26 ± 2°C for seed germination and 22 ± 2°C for early seedling growth; relative humidity at 80 ± 2% and 60 ± 2%, respectively; and light conditions of darkness for germination and 250 ± 5 μmol m$^{-2}$ s$^{-1}$ photosynthetic photon flux density for early seedling growth.

Ten seedlings, upon reaching a height of 25–30 cm, were transplanted in each replicated field plot during the third week of May in both 2022 (May 19) and 2023 (May 21). Plastic-mulched raised beds were prepared with row spacing of 1.6 m and intra-row plant spacing of 60 cm, using a waterwheel transplanter.

Nano-NPK fertilization was administered via drip irrigation over a six-week period, beginning in June and continuing through mid-August in both years. SA applied twice, once in mid-June and again in early July, coinciding with the peak vegetative growth stage.

Irrigation scheduling was based on 75% of the maximum allowable depletion of available soil moisture and was delivered through drip emitters positioned at a 10 cm soil depth. During treatment applications, all irrigation valves were closed except for the specific line delivering the nano-NPK or SA treatments. Treatments were injected into the pressurized drip system, requiring approximately 15 min for injection, followed by an additional 10 min of clean water irrigation to flush the lines. The header line was then uncapped to drain residual solution between treatments.

All other cultural and plant protection practices, including herbicide and fungicide applications, were conducted according to the *Midwest Vegetable Production Guide for Commercial Growers*-ID 56 [40].

**Tomato yield attributes.** Tomato fruits were harvested upon ripening, with harvested fruits graded as marketable or cull (including green), and their respective weights were recorded. Additionally, the average weight of tomatoes and per plant calculation was performed based on randomly selected fifty fresh tomatoes from each replicated plot [28].

## Water- and nutrient-use efficiency

Tomato WUE as well as FUE were calculated [41,42] as follows:

WUE (kg m$^{-3}$) = Total fruit yield (Ton ha$^{-1}$) / irrigation applied (m$^3$ ha$^{-1}$).

FUE (kg kg$^{-1}$) = Total fruit yield (Ton ha$^{-1}$) / NPK fertilizers applied (kg ha$^{-1}$).

                                                                                     

## Soil and water analysis

Composite soil (2.54 cm internal diameter) was collected from 0–20 cm depth using a JMC® stainless-steel soil environmental probe fitted with plastic liners before layout of the field experiment. The soil samples were immediately placed in sealable plastic bags and stored at 4°C until further analysis. A portion of the field-moist soil was air-dried under shade at room temperature (~ 25 °C) for 15 days, ground using an agate mortar and pestle, and passed through a 2 mm sieve prior to analysis.

Soil pH was measured using a glass electrode pH meter (Model 520A, Orion®, Boston, MA, USA). Electrical conductivity was determined in a 1:1 soil-to-distilled water suspension using a conductivity probe. Soil total organic carbon (SOC) and total nitrogen (TN) concentrations were analyzed on finely ground (<125 μm) air-dried samples using a FlashEA 1112 Series CNHS-O dry combustion analyzer®. Bulk density was assessed following the standard core method. Soil particle size distribution was determined using the Bouyoucos hydrometer method, following the oxidation of soil organic matter with 3% hydrogen peroxide and subsequent overnight dispersion in a 10% sodium hexametaphosphate solution. Soil field moisture capacity and permanent wilting point were determined as 0.033 MPa and 1.5 MPa, respectively, using the pressure plate apparatus [43]. Antecedent soil moisture for irrigation scheduling was monitored using a Delta-T soil moisture probe.

Groundwater used for drip irrigation was collected randomly in clean, dry, sterilized plastic bottles. Prior to analysis, samples were filtered through Whatman® No. 42 filter paper. The filtered water samples were analyzed for pH, electrical conductivity, macro- and micronutrients, and heavy metals using standard analytical protocols [28].

## Economic, energy, and environmental assessments

Economic inputs for tomato production were operating costs such as pre-planting and post-planting fertilization, labor, machinery, diesel fuel, agricultural chemicals, sodium salicylate, and Nano Pro solution, water, plastic mulching, drip irrigation system, seed, and cash overhead costs and non-cash overhead costs. Economic outputs were the gross revenues, which were calculated by the total product yields (tomato fruits) and market prices in 2022 and 2023. For calculation of economic benefit, the net return, benefit: cost, and economic productivity were determined [32–33] using the following equations:

$$\text{Net return (\$ ha}^{-1}) = \text{Economic output} - \text{economic input}$$

$$\text{Benefit : cost ratio} = \text{Economic output} \div \text{economic input}$$

$$\text{Economic productivity (kg \$}^{-1}) = \text{Total tomato yield} \div \text{economic input}.$$

Energy inputs of tomato production include energy from plastic mulching, pre- and post-planting fertilizations, Nano Pro solution, sodium salicylate, labor, machinery, diesel fuel, water, drip irrigation system, and seed. Energy outputs of tomatoes consisted of equivalent energy production from fresh tomato fruits. Input and output data were converted into common energy units with the appropriate coefficients of energy equivalence (S1 – S8 Tables in S2 File). Energy use efficiency, net energy, and energy productivity were calculated for energy analysis [44,45], which are as follows:

$$\text{Energy use efficiency or energy rate (\%)} = (\text{Energy output / energy input}) * 100.$$

$$\text{Net energy (MJ ha}^{-1}) = \text{Energy output} - \text{energy input}.$$

$$\text{Energy productivity (kg MJ}^{-1}) = \text{Fresh root, tuber, or grain yield} \div \text{energy input.}$$

Greenhouse gas (GHG) emissions from the tomato production, manufacturing, use and disposing of plastic mulching, machinery, diesel fuel, fertilizers, agricultural chemicals, Nano Pro solution, sodium salicylate, and seeds were estimated (S1 – S8 Tables in S2 File). GHG emissions including carbon dioxide ($CO_2$), methane ($CH_4$), and nitrous oxide ($N_2O$) were weighed as the GHG emission coefficients ($CO_2$-equivalent) according to their global warming potential factors of 1, 25, and 298, respectively, following IPCC's 100-year estimates [44,46]. GHG emissions and GHG intensity of tomato production were calculated [38] as below.

$$\text{Total GHG emission (kg CO}_2 - \text{eq ha}^{-1}) = \sum Q_i \times C_i$$

where $Q_i$ is the quantity of input (i), and $C_i$ indicates the GHG emission coefficients of input (i).

$$\text{GHG intensity (kg CO}_2 \text{ ton of tomato}^{-1}) = \text{Total GHG emission / total tomato yield.}$$

### Statistical analysis

Multivariate statistical analyses were performed using the General Linear Model (GLM) procedure in SAS 9.4° [47]. A two-way analysis of variance (ANOVA) was applied to evaluate the main and interactive effects of nano-NPK fertilization and SA levels (as fixed factors) on processing tomato response variables, with block considered a random effect across years. Year was treated as a repeated factor (additional replication) since no significant interactions with the fixed factors were detected.

The response variables included total, marketable, and cull fruit yields; WUE and FUE; and economic, energy, and environmental performance indicators. Data normality was assessed using the Shapiro–Wilk test, and the homogeneity of variances was verified using Levene's test. Mean separations for all response variables (tomato yield traits and economic parameters) were conducted using least square means, and significant differences were determined with an F-protected Duncan's Multiple Range Test (DMRT) at $\alpha = 0.05$. All graphical outputs were generated using SigmaPlot° 15.

## Results and discussion

### Tomato yield attributes

Tomato yield components were significantly, though variably, influenced by nano-NPK fertilization and SA application, whereas interaction between these factors was not significant (Table 1; Fig 1). Average fruit yield per plant differed among treatments, with nano-NPK applied at 80 and 120 kg ha⁻¹ resulting in slightly higher yields than the conventional NPK control (120 kg ha⁻¹), corresponding to a 14−17% increase compared to 40 kg ha⁻¹ nano-NPK fertilization. Although average fruit weight did not differ significantly among treatments, total fruit yield increased by 2% and 10% with nano-NPK fertilization at 80 and 120 kg ha⁻¹, respectively, but decreased by 33% under 40 kg ha⁻¹ nano-NPK fertilization relative to the control (120 kg ha⁻¹ NPK). Marketable yield increased significantly, by 6% and 17% under nano-NPK fertilizations of 80 and 120 kg ha⁻¹, respectively, compared to the control (Fig 1). Additionally, nano-NPK fertilization at these rates significantly reduced cull fruit yield by 11–23% and increased the marketable yield to cull fruit ratio by 23–43%. In contrast, nano-NPK applied at 40 kg ha⁻¹ resulted in the lowest marketable yield, 33% below the control.

Application of 0.3% SA alone significantly increased total yield by 10% and marketable yield by 15%, decreased cull fruit yield by 8%, and increased marketable yield to cull fruit yield by 25% compared to the control, without affecting other yield traits. An interaction between SA and nano-NPK fertilization was observed, resulting in a significant increase in the marketable yield of processing tomatoes.

Further analyses indicated that both total and marketable tomato yields exhibited significant, non-linear responses to nano-NPK fertilization, regardless of SA application (Figs 2 and 3). Application of 80 kg ha⁻¹ nano-NPK combined

**Table 1. Effects of different rates of nano-NPK fertilization (NF-NPK) and salicylic acid (SA) application on tomato fruit attributes across growing seasons (2022 and 2023). Data are presented as mean values±standard error.**

| SA | NPK | Fruit yield | Aver. fruit | Total fruit | Cull fruit | Marketable fruit |
|---|---|---|---|---|---|---|
| (%) | (kg ha⁻¹) | (kg plant⁻¹) | weight (g) | __ yield (Mg ha⁻¹)__ | | : cull fruit |
| Control | | 2.1$\pm$0.6x≠ | 120.9$\pm$9x | 39.5$\pm$3.0y | 7.5$+$1.9x | 4.6$+$0.6y |
| SA | | 2.2$\pm$0.5x | 117.7$\pm$9x | 43.5$\pm$2.7x | 6.9$\pm$1.9y | 6.1$\pm$0.7x |
| | 120-NPK | 2.1$\pm$0.7ab¥ | 121.2$\pm$8.4a | 43.8$\pm$1.5b | 8.2$\pm$2.4a | 4.8$\pm$0.3c |
| | 40-NFNPK | 1.9$\pm$0.6b | 115.4$\pm$11a | 29.3$\pm$2.3c | 7.3$\pm$1.4ab | 3.1$\pm$0.5d |
| | 80-NFNPK | 2.2$\pm$0.4a | 118.2$\pm$6.1a | 44.6$\pm$3.2b | 6.9$\pm$2ab | 6.2$\pm$0.6ab |
| | 120-NFNPK | 2.3$\pm$0.5a | 122.3$\pm$10.7a | 48.2$\pm$4.3a | 6.3$\pm$1.7b | 7.3$\pm$1.2a |
| **Nano-NPK x SA interaction** | | | | | | |
| Control | 120-NPK | 2.1$\pm$0.5 | 125$\pm$8.2 | 42$\pm$2 | 8.7$\pm$2.5 | 4.2$\pm$0.3 |
| | 40-NFNPK | 1.7$\pm$0.7 | 112.3$\pm$11 | 27.3$\pm$1.7 | 7.2$\pm$1.3 | 2.9$\pm$0.4 |
| | 80-NFNPK | 2.3$\pm$0.7 | 119.7$\pm$5 | 43$\pm$4.1 | 7.2$\pm$1.8 | 5.4$\pm$0.8 |
| | 120-NFNPK | 2.2$\pm$0.5 | 126.6$\pm$12 | 45.6$\pm$4.1 | 7$\pm$1.9 | 6.1$\pm$0.8 |
| SA | 120-NPK | 2.1$\pm$1 | 117.4$\pm$8.7 | 45.7$\pm$1 | 7.7$\pm$2.3 | 5.5$\pm$0.2 |
| | 40-NFNPK | 2.2$\pm$0.5 | 118.4$\pm$11 | 31.4$\pm$2.9 | 7.5$\pm$1.5 | 3.4$\pm$0.6 |
| | 80-NFNPK | 2.1$\pm$0.2 | 116.8$\pm$7.1 | 46.2$\pm$2.2 | 6.5$\pm$2.2 | 7$\pm$0.4 |
| | 120-NFNPK | 2.4$\pm$0.4 | 118$\pm$9.4 | 50.8$\pm$4.5 | 5.7$\pm$1.5 | 8.5$\pm$1.6 |
| *Probability$\geq$F* | | | | | | |
| Nano-NPK x SA | 0.426 | 0.117 | 0.823 | 0.691 | 0.581 | |

≠ Means separated by same lower-case letter (x and y) under each column were not significantly different between salicylic acid treatments at $p\geq$0.05.

¥ Means separated by same lower-case letter (a, b, c, and d) under each column were not significantly different among nano-NPK fertilization rates at $p\geq$0.05.

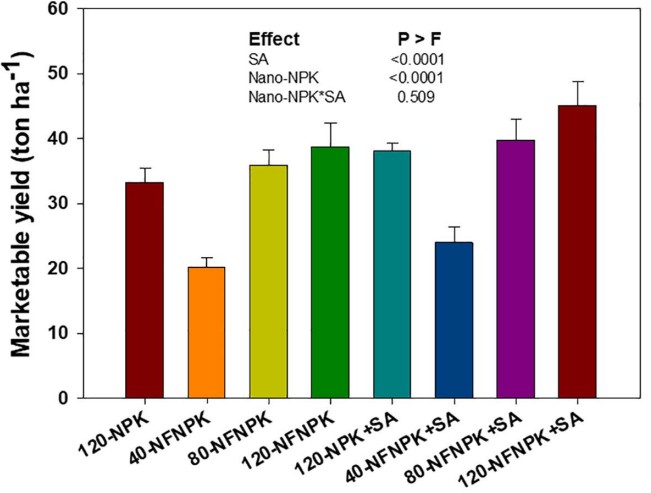

**Fig 1. Effects of varying rates of nano-NPK fertilization and salicylic acid (SA) application on the marketable yield of processing tomato under drip-irrigated plasticulture, averaged across the 2022 and 2023 growing seasons.** Data are presented as mean values±standard error.

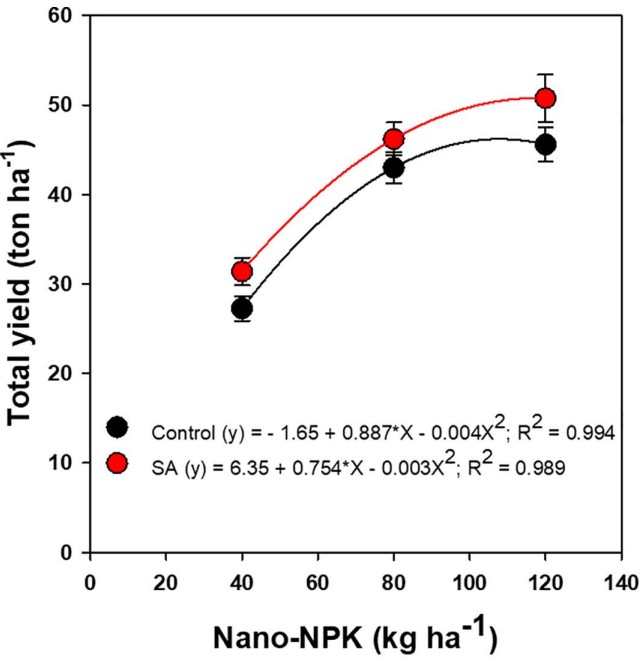

**Fig 2. Regression of total yield of processing tomato on nano-NPK fertilization and salicylic acid (SA) application under drip-irrigated plasti-culture, averaged across the 2022 and 2023 growing seasons.** Data are presented as mean values ± standard error.

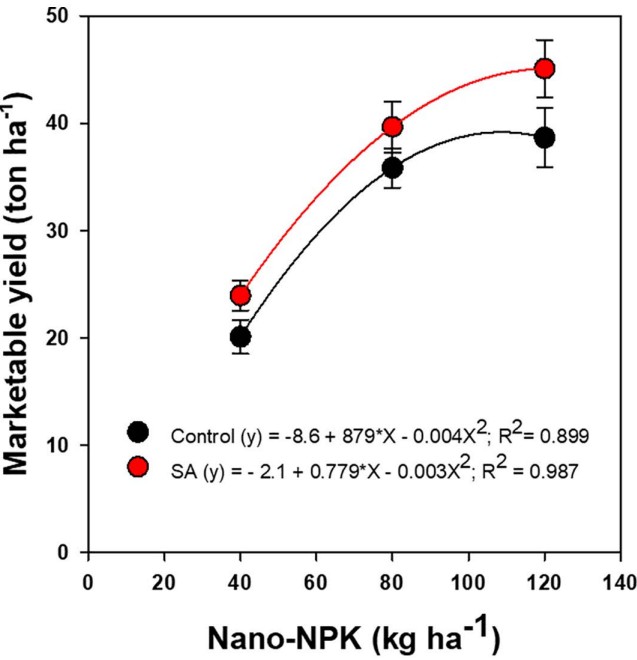

**Fig 3. Regression of marketable yield of processing tomato on nano-NPK fertilization and salicylic acid (SA) application under drip-irrigated plasticulture, averaged across the 2022 and 2023 growing seasons.** Data are presented as mean values ± standard error.

with 0.3% SA yielded total fruit outputs comparable to or exceeding those obtained with 120 kg ha⁻¹ nano-NPK alone by accounting 99% of the variability in total yield (Fig 2). Similarly, marketable yield demonstrated a pronounced non-linear response to varying nano-NPK application rates (Fig 3). Notably, the combination of 80 kg ha⁻¹ nano-NPK fertilization with 0.3% SA produced yields that were statistically comparable to, or even exceeded, those achieved with 120 kg ha⁻¹ nano-NPK alone. This combination accounted for 90–98% of the observed variability in marketable yield, highlighting its efficiency in optimizing crop productivity while potentially reducing fertilizer input.

The significant increases in both total and marketable fruit yields of processing tomato, along with a reduction in cull fruit production at 80 and 120 kg ha⁻¹ nano-enabled NPK fertilization, can be attributed to a lower proportion of cull fruits. This enhancement in yield was primarily due to improve nutrient availability, uptake efficiency, and utilization by the plants, which in turn increase photosynthesis and the translocation of assimilates to the fruits [15,17]. Nanomaterials, when combined with chemical fertilizers, function as efficient carriers, facilitating nutrient transport through root nanopores, increasing uptake, and boosting metabolic function and yield potential [15,16,48]. These findings align with previous research demonstrating increased vegetable crop productivity, including tomatoes, following nano-fortified chemical fertilizer applications [15,16,28]. Notably, it was observed that there was a 25% increase in marketable tomato yield when Fe nanoparticles replaced chelated Fe in commercial NPK formulations [16].

Furthermore, the application of 0.3% SA significantly enhanced total and marketable yields, as expected, due to its ability to mitigate abiotic and biotic stress, improve plant vigor, and regulate physiological and biochemical processes related to water and nutrient uptake, and fruit development, thereby boosting yield potential [21,26,28,49].

Although both nano-NPK and SA individually enhance the growth and yield of tomato, the significant interaction between them on marketable yield arises from their additive yet non-linear effects on physiological and metabolic processes. Specifically, the combination of 80 kg ha⁻¹ nano-NPK with 0.3% SA produced fruit yields comparable to, or even exceeding, those produced with 120 kg ha⁻¹ of nano-NPK fertilization or conventional 120 kg ha⁻¹ NPK control. These results indicate that integrating SA with reduced nano-fertilizer doses can optimize marketable yield while potentially lowering production costs and reducing environmental impacts, highlighting a sustainable and efficient fertilization strategy for tomato cultivation.

## Water- and fertilizer-use efficiency of tomato

Tomato plants WUE were significantly affected by nano-NPK fertilization and SA application, as well as by their interaction (Fig 4). The application of nano-NPK at rates of 80 and 120 kg ha⁻¹ resulted in notable increases in WUE by approximately 13% and 34%, respectively, compared with the conventional NPK fertilization rate of 120 kg ha⁻¹. Likewise, application of SA at a concentration of 0.3% significantly enhanced WUE by 11% relative to the untreated control. Moreover, the combined application of nano-NPK and SA exhibited significant interactive effect, further improving the WUE of processing tomato cultivation.

The FUE of tomato plants was significantly influenced by both nano-NPK fertilization and SA application, as well as by their interaction (Fig 5). Increasing the rate of nano-NPK fertilizer resulted in a progressive decline in FUE, decreasing from 156% at the lowest nano-NPK level (40 kg ha⁻¹) to 74% at the highest (120 kg ha⁻¹). Despite this decline, all nano-NPK treatments maintained higher FUE values, by an average of 61%, compared with the conventional NPK fertilization rate of 120 kg ha⁻¹. Furthermore, application of SA at 0.3% significantly enhanced FUE by approximately 23% relative to untreated plants. The significant interaction observed between nano-NPK fertilization and SA application indicates a synergistic effect on the nutrient use efficiency of tomato plants.

Regression analysis showed a statistically significant, non-linear enhancement in WUE in response to increasing rates of nano-NPK fertilization, irrespective of SA application (Fig 6). The fitted model indicated that WUE of tomato plants followed a curvilinear trajectory, with a diminishing response beyond the optimal nano-NPK rate. Interestingly, the combined treatment of 80 kg ha⁻¹ nano-NPK with 0.3% SA achieved marketable yields that were statistically comparable to those obtained

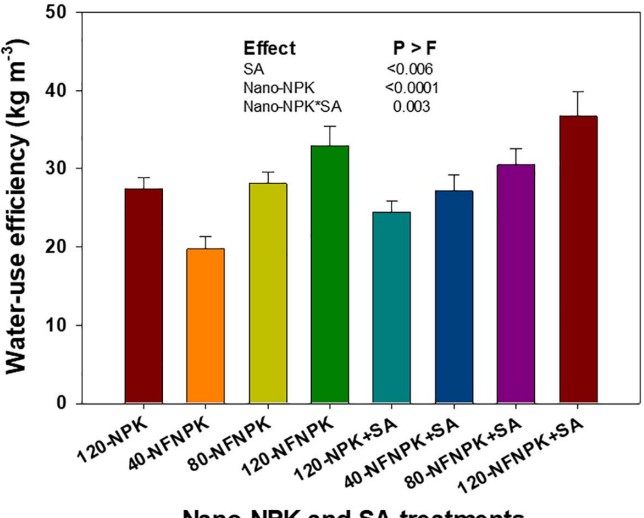

**Fig 4. Effects of nano-NPK fertilization and salicylic acid (SA) application on water-use efficiency of processing tomato under drip irrigated plasticulture, averaged across growing seasons (2022 and 2023).** Data are presented as mean values ± standard error.

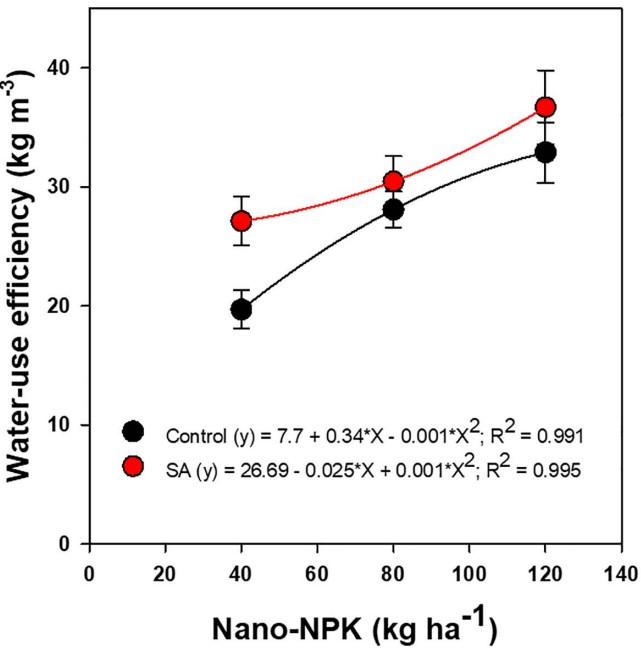

**Fig 5. Effects of nano-NPK fertilization and salicylic acid (SA) application on fertilizer-use efficiency of processing tomato under drip irrigated plasticulture across growing seasons (2022 and 2023).** Data are presented as mean values ± standard error.

with 120 kg ha[-1] nano-NPK applied without SA. This model accounted for approximately 99% of the observed variation in WUE, indicating strong predictive relationship among nano-NPK rate, SA application, and WUE performance. Furthermore, treatments involving nano-NPK at either 80 or 120 kg ha[-1] and SA at 0.3%, whether applied singly or in combination, consistently equaled or surpassed the WUE achieved under the conventional NPK fertilization rate of 120 kg ha[-1].

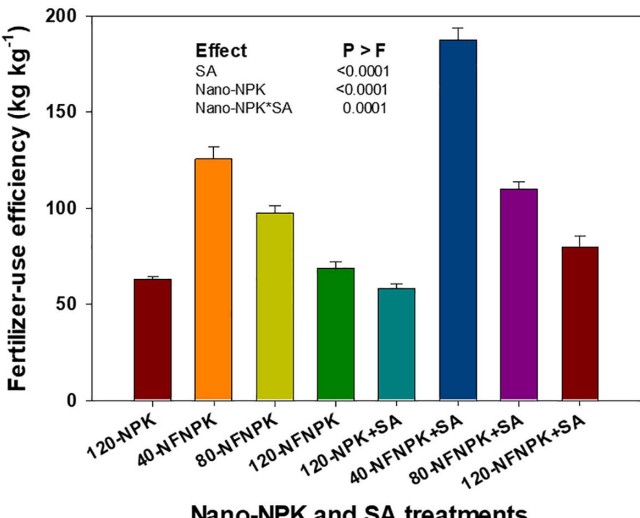

**Fig 6. Regression of water-use efficiency of processing tomato as affected by nano-NPK fertilization and salicylic acid (SA) under drip-irrigated plasticulture, averaged across the 2022 and 2023 growing seasons.** Data are presented as mean values ± standard error.

Regression analysis showed that the FUE of tomato plants declined significantly and non-linearly with increasing nano-NPK application rates, regardless of SA treatment (Fig 7). This pattern indicates that beyond an optimal nutrient threshold, further increases in nano-NPK supply reduce nutrient utilization efficiency. Notably, the combined application of 80 kg ha⁻¹ nano-NPK with 0.3% SA produced marketable yields statistically comparable to those obtained with 120 kg ha⁻¹ nano-NPK without SA. This finding suggests a synergistic role of SA in enhancing nutrient uptake and utilization efficiency at moderate nano-fertilizer doses. Moreover, treatments receiving 120 kg ha⁻¹ nano-NPK, either with or without 0.3% SA, achieved FUE values equivalent to or greater than those observed under conventional NPK fertilization, underscoring the superior performance and efficiency of nano-formulated fertilizers in tomato cultivation.

Improved WUE and FUE observed with nano-NPK fertilization, particularly at 80 and 120 kg ha⁻¹, can be attributed to improved nutrient and water uptake. This effect is facilitated by the nanoporous structure of plant roots, which enhances the rapid absorption and translocation of nutrient-bound nanomaterials [8,15,16]. However, diminishing returns in FUE at higher nano-NPK rates may arise from fertilizer inefficiencies, where excess nutrients fail to contribute to further yield gains [50]. Similarly, previous studies have reported that, although conventional chemical fertilization can increase crop yield and income stability, excessive application does not provide additional yield benefits and may exacerbate environmental pollution [28,50,51].

Furthermore, this study demonstrates a synergistic interaction between nano-fertilization and SA application, particularly the combination of 80 kg ha⁻¹ nano-NPK with 0.3% SA. This suggests that their combined application can optimize WUE and FUE, enhance tomato production under limited water availability, and potentially reduce fertilizer input without compromising productivity. These findings underscore a cost-effective and environmentally sustainable alternative to conventional high-input NPK fertilization practices.

## Economic viability of tomato production

Nano-NPK fertilization and SA application variably affected the average total economic inputs (costs), outputs (gross revenues), benefit: cost ratio, and net returns (profits) for tomato cultivation, with no significant interactions except for

**Fig 7. Regression of fertilizer-use of processing tomato as affected by nano-NPK fertilization and salicylic acid (SA) under drip-irrigated plasticulture, averaged across the 2022 and 2023 growing seasons.** Data are presented as mean values ± standard error.

input costs (Table 2; S1-S8 Tables in S2 File). The average input cost was highest ($7,117) for the 120 kg ha⁻¹ nano-NPK, followed by $6,7657 for 80 kg ha⁻¹, and $6,445 for 40 kg ha⁻¹ nano-NPK fertilization, and was statistically like the conventional 120 kg ha⁻¹ NPK treatment. This indicates only a marginal increase in production costs with higher nano-NPK fertilization rates (120 kg ha⁻¹) but a notable reduction at lower rates (80 and 40 kg ha⁻¹) compared with conventional NPK fertilization.

In contrast, the average economic outputs were significantly higher for 120 kg ha⁻¹ nano-NPK ($9,105), followed by 80 kg ha⁻¹ nano-NPK ($8,448) and 120 kg ha⁻¹ NPK (8,278), compared with 40 kg ha⁻¹ nano-NPK ($5,542). Consequently, net returns were greater for 80 and 120 kg ha⁻¹ nano-NPK ($1,988 and $1,683, respectively) than for 120 kg ha⁻¹ NPK ($1,193), whereas 40 kg ha⁻¹ nano-NPK resulted in a negative return (−$903).

The benefit: cost ratio was higher under nano-NPK fertilization (1.25–1.26) except at 40 kg ha⁻¹ (0.8), compared with the conventional NPK treatment (1.16). Similarly, economic productivity of tomato production followed a comparable trend to the benefit: cost ratios.

The application of 0.3% SA had a significant positive impact on the economic performance of tomato cultivation. Specifically, it increased total income by $779 ha⁻¹, enhanced net return by $794 ha⁻¹, improved the benefit: cost ratio by 0.11, and raised economic productivity by 0.7 kg $⁻¹. These improvements indicate that SA applications can effectively boost profitability and resource-use efficiency in tomato cultivation. However, the application of SA did not significantly alter the total production costs, suggesting that the observed economic gains were primarily due to increased yield or quality rather than cost reductions. Moreover, there was a significant interaction effect between nano-NPK fertilization and SA application on total production costs.

Despite only marginal differences in input costs, economic output of tomatoes significantly increased by the application of nano-NPK fertilizers and SA. The highest economic returns were observed with 80 and 120 kg ha⁻¹ nano-NPK, suggesting a proactive fertilization effect on tomato productivity [52,53]. In contrast, the 40 kg ha⁻¹ nano-NPK treatment

**Table 2. Effects of different rates of nano-NPK fertilization (NF-NPK) and salicylic acid (SA) application on economic viability including total input, total output, net return, benefit: coast ratio, and economic productivity of tomato cultivation across growing seasons (2022 and 2023). Data are presented as mean values ± standard error.**

| SA | NPK | Total output | Total input | Net return | Benefit: | Econ. Prod. |
|---|---|---|---|---|---|---|
| (%) | (kg ha⁻¹) | ($ ha⁻¹) | | | cost ratio | (kg $⁻¹) |
| Control | | 7454+88y≠ | 6860+0.0x | 593+44y | 1.08+0.05y | 5.7+0.3y |
| SA | | 8233+64x | 6846+0.0x | 1387+32x | 1.19+0.03x | 6.4+0.2x |
| | 120-NPK | 8278+33b¥ | 7085+0.0a | 1193+16b | 1.16+0.03a | 6.2+0.1a |
| | 40-NFNPK | 5542+65c | 6445+0.0c | -903+32c | 0.86+0.03b | 4.6+0.2b |
| | 80-NFNPK | 8448+94b | 6765+0.0b | 1683+47a | 1.25+0.04a | 6.7+0.3a |
| | 120-NFNPK | 9105+113a | 7117+0.0a | 1988+56a | 1.26+0.07a | 6.8+0.3a |
| **Nano-NPK x SA interaction** | | | | | | |
| Control | 120-NPK | 7919+62 | 7077+0.0 | 825+31 | 1.13+0.05 | 5.9+0.1 |
| | 40-NFNPK | 5150+7 | 6437+0.0 | −1303+18 | 0.8+0.07 | 4.3+0.1 |
| | 80-NFNPK | 8127+135 | 6757+0.0 | 1353+67 | 1.2+0.08 | 6.4+0.5 |
| | 120-NFNPK | 8618+120 | 7114+0.0 | 1498+60 | 1.2+0.08 | 6.5+0.3 |
| SA | 120-NPK | 8637+151 | 7094+0.0 | 1560+15 | 1.2+0.1 | 6.5+0.1 |
| | 40-NFNPK | 5935+93 | 6454+0.0 | −503+46 | 0.93+0.05 | 5+0.3 |
| | 80-NFNPK | 8770+54 | 6774+0.0 | 2012+27 | 1.3+0.08 | 6.9+0.2 |
| | 120-NFNPK | 9592+106 | 7120+0.0 | 2478+53 | 1.33+0.05 | 7.2+0.3 |
| *Probability>F* | | | | | | |
| Nano-NPK x SA | 0.803 | <0.001 | 0.817 | 0.726 | 0.913 | |

≠ Means separated by same lower-case letter (x and y) under each column were not significantly different between salicylic acid treatments at p>0.05.

¥ Means separated by same lower-case letter (a, b, c, and d) under each column were not significantly different among nano-NPK fertilization rates at p>0.05.

resulted in negative returns, indicating that this fertilization level was insufficient to meet tomato nutrient demands. The corresponding benefit: cost ratios support previous findings that nano-fertilizers enhance economic returns by improving WUE and FUE and reducing losses [54–56]. The improved economic traits of tomatoes following SA application indicate that SA plays a crucial role in enhancing both the quality and marketable yield of the crop. Furthermore, a significant interaction between nano-NPK fertilization and SA application on total production costs demonstrates that integrated nutrient management influences overall expenditure, underscoring the importance of combining advanced fertilization techniques with plant growth biostimulants to optimize both cost efficiency and profitability.

Given its market value and responsiveness to nutrient management, cultivating processing tomato with 80 kg ha⁻¹ nano-NPK combined with 0.3% SA offers an economically advantageous alternative to conventional fertilization practices and other traditional crops [57].

### Energy efficiency of tomato cultivation

Tomato cultivation under nano-NPK fertilization and SA application exhibited a variable trade-off between energy input and output across all treatments, with no significant interaction effects (Table 3; S1–S8 Tables in S2 File). Total energy input was significantly higher (by >5%) under 120 kg ha⁻¹ nano-NPK compared to the conventional 120 kg ha⁻¹ NPK treatment, whereas it was similar or lower (by >7%) under 80 and 40 kg ha⁻¹ nano-NPK. In contrast, energy output was substantially greater with nano-NPK, reaching 38,485 MJ ha⁻¹ at 120 kg ha⁻¹ and 34,468 MJ ha⁻¹ at 80 kg ha⁻¹, compared to the conventional NPK treatment, while the lowest output (25,075 MJ ha⁻¹) was recorded at 40 kg ha⁻¹ nano-NPK. Overall, nano-NPK fertilization at 80 and 120 kg ha⁻¹ achieved 4–14% higher energy output relative to the conventional NPK treatment.

**Table 3. Effects of different rates of nano-NPK fertilization (NF-NPK) and salicylic acid (SA) application on energy efficiency and environmental compatibility including energy input and output, energy-use efficiency, energy productivity, net energy, greenhouse gas (GHG) emission, and greenhouse gas ratio of tomato cultivation across growing seasons (2022 and 2023). Data are presented as mean values ± standard error.**

| SA (%) | NPK (kg ha⁻¹) | Energy input (MJ ha⁻¹) | Energy output (MJ ha⁻¹) | Energy-use efficiency (%) | Energy productivity (kg MJ⁻¹) | Net energy (MJ ha⁻¹) | GHG emission (kg CO₂eq ha⁻¹) | GHG intensity (kg CO₂ ton⁻¹) |
|---|---|---|---|---|---|---|---|---|
| Control | | 72529±500x≠ | 30639±1043y | 39.3±4.3y | 0.54±0.04x | -42881±956x | 1802±40x | 48.7±5.7x |
| SA | | 75015±438x | 34888±565x | 45.2±2.0x | 0.56±0.01x | -40880±254x | 1863±35x | 45±2.5x |
| | 120-NPK | 74553±374ab¥ | 33025±1098b | 41±5.3b | 0.49±0.06c | -41946±158ab | 1824±40a | 42.1±6.3b |
| | 40-NFNPK | 69002±751b | 25075±859c | 35.1±2.8c | 0.61±0.02a | -44640±145b | 1753±55a | 58.8±4.6a |
| | 80-NFNPK | 72760±376ab | 34468±778b | 45.8±3.3a | 0.58±0.02ab | -39356±384a | 1822±28a | 44.6±4.4b |
| | 120-NFNPK | 78772±376a | 38485±481a | 47.2±1.3a | 0.54±0.01bc | -41579±373ab | 1932±28a | 41.8±1.0b |
| **Nano-NPK x SA interaction** | | | | | | | | |
| Control | 120-NPK | 69583±497 | 29490±2032 | 36.3±10.1 | 0.45±0.13 | −40733±298 | 1703±49 | 41.4±12.1 |
| | 40-NFNPK | 69002±751 | 23360±795 | 33±2.6 | 0.61±0.03 | −46102±78 | 1753±55 | 62.2±4.6 |
| | 80-NFNPK | 72760±376 | 33175±838 | 43.5±3.2 | 0.58±0.02 | −41032±387 | 1822±28 | 47±4.7 |
| | 120-NFNPK | 78772±376 | 36530±507 | 44.6±1.4 | 0.54±0.01 | −43657±403 | 1932±28 | 44.2±1.2 |
| SA | 120-NPK | 79523±250 | 36560±163 | 45.7±0.4 | 0.53±0.00 | −43158±78 | 1946±31 | 42.9±0.4 |
| | 40-NFNPK | 69003±751 | 26790±922 | 37.3±3.1 | 0.61±0.02 | −43178±212 | 1753±55 | 55.3±4.6 |
| | 80-NFNPK | 72760±376 | 35760±717 | 48.1±3.4 | 0.58±0.02 | −37680±382 | 1822±28 | 42.3±4.2 |
| | 120-NFNPK | 78772±376 | 40440±456 | 49.9±1.2 | 0.54±0.01 | −39502±344 | 1932±28 | 39.5±0.8 |
| *Probability > F* | | | | | | | | |
| Nano-NPK x SA | 0.438 | 0.668 | 0.593 | 0.263 | 0.429 | 0.426 | 0.449 | |

≠ Means separated by same lower-case letter (x and y) under each column were not significantly different between SA treatments at p≥0.05.

¥ Means separated by same lower-case letter (a, b, c, and d) under each column were not significantly different among nano-NPK fertilization rates at p≥0.05.

Energy use efficiency, a critical indicator of system sustainability, was 4–6% higher under nano-NPK fertilization at 80 and 120 kg ha⁻¹ than under conventional rate of NPK, although non- significant difference was observed between the two nano-NPK rates. Similarly, energy productivity was comparable among nano-NPK rates but remained significantly higher (by 0.09–0.12 kg MJ⁻¹) than that of conventional NPK fertilization. Despite all treatments exhibiting a negative energy balance, nano-NPK fertilization, particularly at 80 kg ha⁻¹, mitigated net negative energy use.

The application of 0.3% SA did not significantly alter total energy input, energy productivity, or net energy use; however, it significantly improved energy output and energy use efficiency by 4,249 MJ ha⁻¹ and 6%, respectively. Across treatments, total energy input exceeded output, leading to a net negative energy balance, which is characteristic of high-input, intensive horticultural systems [58]. Among input components, plastic mulch contributed the highest share of total energy consumption, followed by NPK fertilizers, diesel fuel, and herbicides, consistent with findings from similar energy audits in vegetable cropping systems [59,60].

The observed improvements in energy output and energy use efficiency with nano-NPK fertilization (particularly at 80 and 120 kg ha⁻¹) and 0.3% SA application can be attributed to enhanced nutrient use efficiency and reduced nutrient losses typically associated with nano-fertilizer technology. Moreover, the positive influence of SA stems from its role in regulating plant physiological responses and resource utilization efficiency. These results corroborate previous studies demonstrating that nano-fertilization and SA enhance crop performance and resource optimization, thereby offering a promising strategy to improve the energy sustainability of tomato cultivation systems [28,61].

## Environmental compatibility of tomato cultivation

Environmental compatibility indices, including total greenhouse gas (GHG) emissions and GHG intensity (expressed as ratio of kg GHG emitted per ton of marketable yield), were variably affected by nano-NPK fertilization, SA application, or their interaction (Table 3; S1–S8 Tables in S2 File).

Total GHG emissions showed a non-significant increase at higher nano-NPK fertilization rates (80 and 120 kg ha$^{-1}$) compared to both conventional NPK (120 kg ha$^{-1}$) and lower nano-NPK rates (40 kg ha$^{-1}$). In contrast, GHG intensity significantly decreased with increasing nano-NPK rates, from 58.8 to 41.8 kg $CO_2$ ton$^{-1}$, values statistically comparable to those observed under conventional NPK fertilization (120 kg ha$^{-1}$). Similarly, SA application resulted in a non-significant rise in total GHG emissions but was associated with a 7% reduction in GHG intensity relative to the control.

These findings indicate that while total GHG emissions may remain similar or slightly increase with higher nano-NPK application rates compared to conventional fertilization, emissions per unit of tomato yield (GHG intensity) tend to decrease, an important metric for sustainable crop production. The application of SA further contributed to reduced GHG intensity, though the interaction between nano-NPK and SA treatments was not statistically significant.

Variations in total GHG emissions across treatments are primarily attributed to differences in fertilizer application rates, consistent with prior studies identifying fertilizer inputs, particularly nitrogen, as major contributors to GHG emissions in agricultural systems [62]. Other research has also highlighted that optimizing inputs such as diesel fuel and nitrogen can reduce GHG emissions [31,63].

Overall, these results emphasize the importance of precision nutrient management in minimizing the environmental footprint of crop production. Nano-enabled NPK fertilizers, when applied at optimized rates, especially 80 kg ha$^{-1}$, appear to maintain tomato yield while reducing GHG emissions. This supports the broader conclusion that avoiding excessive use of chemical fertilizers can prevent trade-offs between high productivity and environmental sustainability [18,31,63].

## Conclusions

The combined or individual application of nano-NPK and SA provides significant agronomic, economic, and environmental advantages for processing tomato cultivation. Nano-NPK fertilization markedly enhanced yield components, WUE, and FUE), while reducing cull fruit and improving marketable yield. Among all treatments, nano-NPK at 80 kg ha$^{-1}$ combined with 0.3% SA proved most effective and sustainable. This combination produced total and marketable yields comparable to or exceeding those obtained with 120 kg ha$^{-1}$ nano-NPK or conventional NPK, while ensuring efficient nutrient and water utilization. Concurrently, SA enhanced plant resilience, further contributing to improved tomato productivity.

From a resource-use perspective, both WUE and FUE showed significant non-linear increases, confirming the synergistic role of nano-fertilizers and SA in improving input efficiency and yield stability. The 80 kg ha$^{-1}$ nano-NPK + 0.3% SA treatment achieved WUE and FUE values comparable to or greater than those under 120 kg ha$^{-1}$ conventional NPK, indicating potential fertilizer savings of up to 33% without economic yield, an important consideration for resource-limited or water-scarce systems. Economically, the 80 kg ha$^{-1}$ nano-NPK treatment recorded the highest benefit: cost ratio (1.26) and net return ($1,988 ha$^{-1}$), outperforming conventional NPK treatment. SA further enhanced profitability by boosting yield without increasing production costs.

In terms of sustainability, applying nano-enabled NPK fertilizer, particularly at 80 kg ha$^{-1}$, enhanced energy output and efficiency by 4–6% and reduced GHG intensity. Therefore, integrating 80 kg ha$^{-1}$ nano-NPK with 0.3% SA offers a practical, cost-effective, and environmentally sustainable approach for the intensified production of processing tomatoes.

## Supporting information

**S1 File. Minimum data set – Supplementary.**
(XLSX)

**S2 File. Supporting tables.**

(DOCX)

## Acknowledgments

We are thankful to Professor Gary Gao for allowing us to use his laboratory facilities and Thomas Harker, Research Associate, Wayne Lewis, Farm manager, and undergraduate students for their continuous support to planting, monitoring, data collection, and harvesting tomatoes at CFAES OSU South Centers.

## Author contributions

**Conceptualization:** Khandakar Rafiq Islam, Brad Bergefurd.

**Data curation:** Zhenhao Guan, Khandakar Rafiq Islam, Arifur Rahman.

**Formal analysis:** Zhenhao Guan, Arifur Rahman.

**Funding acquisition:** Khandakar Rafiq Islam, Brad Bergefurd.

**Methodology:** Arifur Rahman.

**Writing – original draft:** Khandakar Rafiq Islam.

**Writing – review & editing:** Khandakar Rafiq Islam.

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
