## [Decision Letter · Decision Letter 0]

29 Sep 2025

PONE-D-25-44279Improving Tomato Yield and Production Economics with Nano-NPK Fertilization and Salicylic Acid ChemigationPLOS ONE

Dear Dr. Islam,

Thank you for submitting your manuscript to PLOS ONE. After careful consideration, we feel that it has merit but does not fully meet PLOS ONE’s publication criteria as it currently stands. Therefore, we invite you to submit a revised version of the manuscript that addresses the points raised during the review process.

If applicable, we recommend that you deposit your laboratory protocols in protocols.io to enhance the reproducibility of your results. Protocols.io assigns your protocol its own identifier (DOI) so that it can be cited independently in the future. For instructions see: https://journals.plos.org/plosone/s/submission-guidelines#loc-laboratory-protocols. Additionally, PLOS ONE offers an option for publishing peer-reviewed Lab Protocol articles, which describe protocols hosted on protocols.io. Read more information on sharing protocols at . Additionally, PLOS ONE offers an option for publishing peer-reviewed Lab Protocol articles, which describe protocols hosted on protocols.io. Read more information on sharing protocols at https://plos.org/protocols?utm_medium=editorial-email&utm_source=authorletters&utm_campaign=protocols..

We look forward to receiving your revised manuscript.

Kind regards,

Somayeh Soltani-Gerdefaramarzi, Ph. D.

Academic Editor

PLOS ONE

Journal Requirements:

“The financial support provided by the Ohio Department of Agriculture via USDA Specialty Crops Block grant (2021-2024)”

“N/A”

6. We note that your Data Availability Statement is currently as follows: All relevant data are within the manuscript and in Supporting Information files.

Reviewers' comments:

Reviewer's Responses to Questions

**Comments to the Author**

1. Is the manuscript technically sound, and do the data support the conclusions?

Reviewer #1: Yes

Reviewer #2: Yes

2. Has the statistical analysis been performed appropriately and rigorously? 

Reviewer #1: Yes

Reviewer #2: Yes

3. Have the authors made all data underlying the findings in their manuscript fully available?

Reviewer #1: Yes

Reviewer #2: No

4. Is the manuscript presented in an intelligible fashion and written in standard English?

Reviewer #1: Yes

Reviewer #2: Yes

5. Review Comments to the Author

Reviewer #1: Abstract

1- Statistical method using in the study has to be stated in the Abstract section;

2- What were the indices for calculating the marketable yield?

Introduction

For optimal management of plant nutrition in tomato, is needed to present some literature reviews, for example for the second paragraph of the Introduction section "lines 69-77", you may use the following work for the literature reviews:

• Optimal management of plant nutrition in tomato (Lycopersicon esculent Mill) by using biologic, organic and inorganic fertilizers. Journal of Plant Nutrition, 46(8): 1560-1579. https://doi.org/10.1080/01904167.2022.2092511

For the lines "103-111", about the importance, functions and critical points to use Salicylic Acid in agriculture, there are some relevant and more up-to-date works to improve the literature reviews of this section; two samples are listed as follows:

1. Effect of Foliar Application of Silicon and Salicylic Acid on Regulation of Yield and Nutritional Responses of Greenhouse Cucumber Under High Temperature. Journal of Plant Growth Regulation, 41(5): 1978-1988. https://doi.org/10.1007/s00344-021-10562-5

2. Salicylic Acid and Biochar-Biofertilizer Improve Soil Fertility, Drought Tolerance, and Fig Yield in a Semi-Arid Region. Journal of Soil Science and Plant Nutrition, pp.1-15. https://doi.org/10.1007/s42729-025-02609-3

The last paragraph of the introduction section, you used the term of "low fertility soils", while this term was not mentioned in the abstract, in the title, please clarify, what it does mean and it was based on which indices;

Materials and Methods

In the manuscript you used "low fertility soils" but the needed information to support this term was missed in the Materials and Methods section;

The software using for statistical analysis has no mention in the Materials and Methods section;

The cultivar has to be stated in the manuscript;

Results

Please use the figures and images with higher resolution;

Conclusion

What is the applied conclusion of this research for farmers and experts? Please state this matter in the conclusion section

Reviewer #2: The study’s integration of agronomic, economic, energy, and environmental assessments provides a holistic perspective on sustainable tomato production.

While the manuscript is scientifically sound and offers valuable insights, several aspects could be improved to strengthen clarity, rigor, and broader applicability.

6. PLOS authors have the option to publish the peer review history of their article (what does this mean?). If published, this will include your full peer review and any attached files.). If published, this will include your full peer review and any attached files.

.

Reviewer #1: No

Reviewer #2: No

While revising your submission, please upload your figure files to the Preflight Analysis and Conversion Engine (PACE) digital diagnostic tool, https://pacev2.apexcovantage.com/. PACE helps ensure that figures meet PLOS requirements. To use PACE, you must first register as a user. Registration is free. Then, login and navigate to the UPLOAD tab, where you will find detailed instructions on how to use the tool. If you encounter any issues or have any questions when using PACE, please email PLOS at . PACE helps ensure that figures meet PLOS requirements. To use PACE, you must first register as a user. Registration is free. Then, login and navigate to the UPLOAD tab, where you will find detailed instructions on how to use the tool. If you encounter any issues or have any questions when using PACE, please email PLOS at figures@plos.org. Please note that Supporting Information files do not need this step.. Please note that Supporting Information files do not need this step.

---

## [Author Response · Author response to Decision Letter 1]

20 Nov 2025

In-depth information was added in Response to reviewers' and editor's comments file

---

## [Decision Letter · Decision Letter 1]

13 Apr 2026

Improving Tomato Yield and Production Economics with Nano-NPK Fertilization and Salicylic Acid Chemigation

PONE-D-25-44279R1

Dear Dr. Islam,

We’re pleased to inform you that your manuscript has been judged scientifically suitable for publication and will be formally accepted for publication once it meets all outstanding technical requirements.

An invoice will be generated when your article is formally accepted. Please note, if your institution has a publishing partnership with PLOS and your article meets the relevant criteria, all or part of your publication costs will be covered. Please make sure your user information is up-to-date by logging into Editorial Manager at Editorial Manager® and clicking the ‘Update My Information' link at the top of the page. For questions related to billing, please contact  and clicking the ‘Update My Information' link at the top of the page. For questions related to billing, please contact billing support..

Kind regards,

Somayeh Soltani-Gerdefaramarzi, Ph. D.

Academic Editor

PLOS One

Additional Editor Comments (optional):

Reviewers' comments:

Reviewer's Responses to Questions

**Comments to the Author**

1. If the authors have adequately addressed your comments raised in a previous round of review and you feel that this manuscript is now acceptable for publication, you may indicate that here to bypass the “Comments to the Author” section, enter your conflict of interest statement in the “Confidential to Editor” section, and submit your "Accept" recommendation.

Reviewer #1: All comments have been addressed

2. Is the manuscript technically sound, and do the data support the conclusions?

Reviewer #1: Yes

3. Has the statistical analysis been performed appropriately and rigorously? 

Reviewer #1: Yes

4. Have the authors made all data underlying the findings in their manuscript fully available?

Reviewer #1: Yes

5. Is the manuscript presented in an intelligible fashion and written in standard English?

Reviewer #1: Yes

6. Review Comments to the Author

Reviewer #1: Dear Authors,

Greetings and regards

Your revised manuscript was carefully checked, I want to say thank you very much for careful revising the manuscript.

Good luck

7. PLOS authors have the option to publish the peer review history of their article (what does this mean?). If published, this will include your full peer review and any attached files.). If published, this will include your full peer review and any attached files.

.

Reviewer #1: No

---

## [Editor Report · Acceptance letter]

PONE-D-25-44279R1

PLOS One

Dear Dr. Islam,

I'm pleased to inform you that your manuscript has been deemed suitable for publication in PLOS One. Congratulations! Your manuscript is now being handed over to our production team.

Kind regards,

on behalf of

Dr. Somayeh Soltani-Gerdefaramarzi

Academic Editor

PLOS One